

# Use of flow cytometry and stable isotope analysis to determine phytoplankton uptake of wastewater derived ammonium in a nutrient-rich river

Calla M. Schmidt[1], Tamara E. C. Kraus[2], Megan B. Young[3], Carol Kendall[3]

[1]University of San Francisco, 2130 Fulton St, San Francisco, CA, 94117, USA
[2]USGS California Water Science Center, 6000 J Street, Placer Hall, Sacramento CA 95819, USA
[3]USGS National Research Program, Menlo Park, CA, USA

*Correspondence to*: Calla M Schmidt (cischmidt@usfca.edu)

**Abstract.** Anthropogenic alteration of the form and concentration of nitrogen (N) in aquatic ecosystems is widespread. Understanding availability and uptake of different N sources at the base of aquatic food webs is critical to establishment of effective nutrient management programs. Stable isotopes of N ($^{14}$N,$^{15}$N) are often used to trace the sources of N fueling aquatic primary production, but effective use of this approach requires obtaining a reliable isotopic ratio for phytoplankton.

In this study, we tested the use of flow cytometry to isolate phytoplankton from bulk particulate organic matter (POM) in a portion of the Sacramento River, California, during river-scale nutrient manipulation experiments that involved halting wastewater discharges high in ammonium ($NH_4^+$). Field samples were collected using a Lagrangian approach, allowing us to measure changes in phytoplankton N source in the presence and absence of wastewater derived $NH_4^+$. Comparison of $\delta^{15}$N-POM and $\delta^{15}$N-Phytoplankton ($\delta^{15}$N-PHY) revealed that their $\delta^{15}$N values followed broadly similar trends. However, after 3

days of downstream travel in the presence of wastewater treatment plant (WWTP) effluent, $\delta^{15}$N-POM and $\delta^{15}$N-PHY in the Sacramento River differed by as much as 7 ‰. Using a stable isotope mixing model approach, we estimated that in the presence of effluent between 40 and 90 % of phytoplankton-N was derived from $NH_4^+$ after 3 days of downstream transport. An apparent gradual increase over time in the proportion of $NH_4^+$ in the phytoplankton N pool suggests that either very low phytoplankton growth rates resulted in an N turnover time that exceeded the travel time sampled during this study or a

portion of the phytoplankton community continued to access nitrate even in the presence of elevated $NH_4^+$ concentrations.



## 1 Introduction

Anthropogenic nutrient enrichment is impacting aquatic ecosystems globally (Smith 2003). In many aquatic environments anthropogenic N loading from wastewater treatment plants, urea-based fertilizers, animal waste, and aquaculture is shifting the form of N available to phytoplankton from the oxidized form of nitrate ($NO_3^-$), to the reduced form of ammonium ($NH_4^+$)

(Glibert et al., 2016). The form of N accessed by phytoplankton is of concern because it has been linked to changes in phytoplankton abundance and species composition, and may provide advantages to less desirable species of phytoplankton, including cyanobacteria that produce harmful toxins (Sharp et al., 2010; Dugdale et al., 2007; Glibert et al., 2011, Paerl et al., 2014). Given widespread alteration to the form and concentration of N in aquatic ecosystems, understanding the availability and uptake of different N sources at the base of the food web is critical to establishment of effective nutrient

management programs (Paerl et al., 2016).

Natural abundance stable isotope analysis is a powerful tool for tracing nutrient sources because the isotopic composition of primary producers reflects the isotopic composition of their source nutrients. Natural abundance approaches have the advantage of integrating over space and time and allowing measurement *in situ*, thus avoiding artifacts introduced in lab-

based studies (Finlay and Kendall, 2007). Natural abundance techniques can also complement experimental studies that use $^{15}$N-labeled substrates, which typically require short-term measurements in isolated volumes that may not accurately represent field conditions. It is possible to capitalize on the distinctive isotopic signatures of anthropogenic N sources, such as sewage, to trace the transport of N through an ecosystem (McClelland and Valiela 1998; Gartner et al., 2002; Schlacher et al., 2005; DeBruyn and Rassmussen 2010; Pennino et al., 2016). Additionally, stable isotope approaches have been used to

distinguish between forms of dissolved inorganic nitrogen (DIN) fueling primary production, which may be particularly important in settings where anthropogenic activities are altering the dominant available N form (York et al., 2007; Sugimoto et al., 2014; Lehman et al., 2014).

Using natural abundance stable isotope techniques to trace the transfer of different N sources into the base of aquatic food

webs requires obtaining a reliable value for the $\delta^{15}$N of phytoplankton ($\delta^{15}$N-PHY). However, few field measurements of phytoplankton $\delta^{15}$N have been published due, in part, to the difficulty of isolating a pure phytoplankton sample from bulk particulate organic matter (POM), which variously contains a mixture of live and dead phytoplankton, macrophyte detritus, bacteria, terrestrial soil and leaves, and (or) sediment with varying isotopic compositions. One approach to solving this challenge is to estimate $\delta^{15}$N-PHY from $\delta^{15}$N-POM when the carbon to nitrogen atomic ratio (C:N) or the carbon to

chlorophyll-*a* weight ratio (C:Chl-*a* ) of POM indicate dominance by phytoplankton. Because terrestrial plant matter, periphyton, and macrophytes have C:N ratios >10, POM with a C:N ratio near the Redfield ratio (6.6 to 8.3) has been used to identify POM primarily composed of phytoplankton (Redfield 1958; Thorp et al., 1998; Kendall et al., 2001). Similarly, a





ratio of C:Chl-*a* less than 200 has been used to identify POM of algal origin, with C:Chl-*a* values above 200 indicating the presence of significant detrital material (Parsons et al., 1961; Cifuentes et al., 1989; Liu et al., 2007; Miller et al., 2013).

To determine $\delta^{15}$N-PHY in settings where POM contains a mixture of organic matter sources, additional approaches have

been developed ranging from physical separation of phytoplankton from bulk POM by density (Hamilton et al., 2005), to isolation of specific compounds such as chlorophyll (Sachs, et al., 1999) or amino acids (McClelland and Montoya, 2002) for $\delta^{15}$N analysis. More recently, Fawcett et al. (2011) demonstrated the use of flow cytometry to separate phytoplankton from bulk POM for $\delta^{15}$N analysis. Cell sorting by flow cytometry is an encouraging new approach for investigations of phytoplankton N source because it theoretically allows for detailed separation of the bulk POM pool into its constituent parts

(detritus, heterotrophic bacteria, phytoplankton, prokaryotes, etc.) prior to isotopic analysis. For example, Fawcett et al. (2011) were able to distinguish differences in prokaryote and eukaryote phytoplankton access to upwelled $NO_3^-$ in the Sargasso Sea using this approach.

Here we report results of a study that tested application of flow cytometry to isolate phytoplankton from bulk POM prior to

isotopic analysis in the Sacramento River, California, in a portion of the San Francisco Bay Estuary (SFE), where $NH_4^+$ concentrations are elevated by WWTP discharges. The goals of this study were to 1) determine the extent to which $\delta^{15}$N-POM reflects $\delta^{15}$N-PHY in the Sacramento River, and 2) trace the *in situ* movement of WWTP derived $NH_4^+$ into phytoplankton using natural abundance stable isotope techniques. This study was conducted during two river-scale nutrient manipulation experiments when WWTP effluent discharges high in $NH_4^+$ were halted, revealing changes in $\delta^{15}$N-POM and

$\delta^{15}$N-PHY in the presence and absence of effluent. To our knowledge, this is the first application of flow cytometry coupled with natural abundance stable isotope analysis in a highly disturbed freshwater system.

The Sacramento-San Joaquin River Delta forms the landward portion of the San Francisco Bay Estuary (SFE), and freshwater flow into the Delta comes primarily from the Sacramento River (Fig. 1). A long-term decline in primary

productivity has been documented in the SFE (Jassby et al., 2002) with resulting declines in zooplankton forage and pelagic fishes (Sommer et al., 2007). A myriad of factors including changes in flow regime, loss of habitat, introductions of exotic bivalve species, and inputs of contaminants and nutrients are believed to contribute to observed reductions in primary productivity (Jassby and Cloern, 2000; Kimmerer, 2002; Muller-Solger et al., 2002; Jassby, 2008). Nutrient concentrations have been increasing in the SFE over time due to agricultural and urban runoff, and increased WWTP discharges, but

discharge from Sacramento Regional WWTP is the main source of $NH_4^+$ in the upper SFE (Jassby, 2008).

The Sacramento Regional WWTP currently employs secondary treatment that does not include a nitrification step, and thus the majority of N in the final effluent is in the form of $NH_4^+$, with little to no N in the form of $NO_3^-$ or nitrite ($NO_2^-$). The



concentration of $NH_4^+$ in treated effluent ranges from 1700-2400 µM, while $NO_3^-$ concentrations are typically below the WWTP's reported detection limit of <0.7 µM (O'Donnell, 2014). Upstream of the WWTP, the concentration of $NH_4^+$ is commonly <0.4 µM, while concentrations of $NH_4^+$ >5 µM are commonly measured downstream of the effluent input (Kratzer et al. 2001; Foe et al., 2010). $NH_4^+$ discharge from WWTPs is of particular concern in the SFE because several

studies have indicated that elevated concentrations of $NH_4^+$ may be causing changes in phytoplankton species abundance and productivity (Dugdale et al., 2007; Glibert et al., 2011; Dugdale et al., 2012; Parker et al., 2012*b*).

## 2 Methods

### 2.1 Field sampling

This study focused on the 70 km channelized reach of the Sacramento River extending from the city of Sacramento

downstream to Isleton (Fig. 1) where the river enters the more hydrodynamically complicated network of open water, channels, and sloughs called the Cache Slough Complex. The only significant inflow within the study reach is just below the Freeport Bridge where treated effluent from the Sacramento Regional WWTP enters the river. River flows are monitored at two USGS stations located at Freeport Bridge and Walnut Grove (http://waterdata.usgs.gov/usa/nwis).

Field sampling was conducted as part of a larger experiment designed to examine changes in phytoplankton abundance and community composition in the presence and absence of wastewater in the Sacramento River. For details of the field methods employed see Kraus et al., (2017). Briefly, field sampling was conducted using a Lagrangian sampling approach during October 24 to 29, 2013, and May 30 to June 4, 2014 (hereafter referred to as the "October" and "June" experiments). During both October and June, sampling was coordinated with ~20 hour WWTP effluent discharge holds, creating a ~15 km stretch

of effluent-free river to allow comparison of two parcels of river water; one containing effluent high in $NH_4^+$ (+EFF) and one without effluent (-EFF). On both dates, +EFF and -EFF parcels were tracked and sampled for ~80 hours (3.5 days) as they travelled ~70 km downstream (Fig. 1). Water samples were collected from both parcels each day at approximately 2 to 3 hour intervals between 8 am and 5 pm PST. Discrete water samples were collected from 1-meter depth using a 3k Shurflo pump with clear ½" tubing using USGS protocols (USGS, 2006). Samples were pumped into 8-L Teflon Jerri cans and then

transferred into a 20-L churn (USGS, 2006) for subsampling. Subsamples were collected for nutrients, chlorophyll-*a* (Chl-*a*), plankton identification and enumeration, $^{15}N$-labeled nutrient uptake experiments, flow cytometry and stable isotope analysis.

### 2.2 Dissolved nutrients and chlorophyll-*a* concentrations

Nutrient and Chl-*a* concentration analyses were performed at the San Francisco State University Romburg Tiburon Center. Water samples for nutrient analysis were immediately filtered through Whatman GF/F filters using a 50-mL syringe into





either 20 mL HDPE scintillation vials or 50 mL centrifuge tubes, placed on dry ice, and then stored at -20 ºC until analysis. Concentrations of $NO_2^-$ and $NO_3^-$ plus $NO_2^-$ were analyzed independently on a Bran and Luebbe AutoAnalyzer II. Samples for $NH_4^+$ determination were collected separately into 50 mL centrifuge tubes after similar filtration. These samples were also immediately frozen for later analysis by colorimetry using a Hewlett Packard diode array spectrophotometer with a 10

cm path cell length.

Chl-*a* samples were concentrated onto 25 mm, 70 µm, Whatman GF/F filters using a low vacuum (<250 mm Hg). Filters were stored dry at 4 ºC for up to one week. Prior to analysis, Chl-*a* was extracted from the filters in 90 % acetone for 24 h at 4 ºC. Analysis was performed fluorometrically with a Turner Designs Model 10-AU using 10 % hydrochloric acid to correct

for and measure phaeophytin. The fluorometer was calibrated with commercially available Chl-*a* (Turners Designs chlorophyll-*a* standard).

### 2.3 Stable isotope analysis of POM, $NO_3^-$ and $NH_4^+$

Stable isotope analysis of POM, $NO_3^-$ and $NH_4^+$ were performed at the US Geological Survey's Menlo Park Stable Isotope
Laboratory. Nitrogen has two stable isotopes, $^{14}N$ and $^{15}N$, and the relative abundance of $^{14}N$ and $^{15}N$ is expressed as $\delta^{15}N$ (‰) = $[(R_{sample}/R_{standard})-1]$ x 1000, where $R_{sample}$ is the ratio of $^{15}N$ to $^{14}N$ in a sample, and $R_{standard}$ is the ratio of the isotopes in AIR, the recognized reference material for $\delta^{15}N$ values. Replicate samples for isotopic analysis of POM were filtered through pre-combusted GF/F filters, and the filters were frozen at -4°C until analysis. Filters were freeze-dried, ground, and vapor-acidified to remove any carbonate prior to analysis for $\delta^{15}N$ and C:N atomic ratio using a Carlo Erba NA 1500
elemental analyzer connected to a Micromass Optima mass spectrometer. Analytical precision for duplicate analyses of the same POM sample was <0.5‰ for $\delta^{15}N$. Samples for $NO_3^-$ isotopes ($\delta^{15}N$- $NO_3$) were filtered through 0.45-µm nucleopore filters, and the filtrate was kept frozen until analysis using a minor modification of the Sigman et al. (2001) and Casciotti et al. (2002) microbial denitrifier method using a modified Gilson autosampler connected to an IsoPrime mass spectrometer. Analytical precision for sample replicates was 0.3 ‰ for $\delta^{15}N$-$NO_3^-$. Samples for $\delta^{15}N$ analysis of $NH_4^+$ ($\delta^{15}N$-$NH_4^+$) were
prepared using a slightly modified version of the method of Holmes et al. (1998) and analyzed on a Carlo Erba NA 1500 elemental analyzer connected to a Micromass Optima mass spectrometer (Kendall et al., 2001). Analysis of $\delta^{15}N$-$NH_4^+$ was only possible on samples with $NH_4^+$ concentrations greater than 15 µM. Precision for this method based on replicate analyses of samples in this study was <0.4 ‰.

### 2.4 Flow cytometry and $\delta^{15}N$ analysis of sorted phytoplankton

At a subset of stations, samples were collected (*n* = 27) for flow cytometric separation of phytoplankton from bulk POM for N-isotopic analysis using the method described in detail in Fawcett et al. (2011). For each sample, 1 liter of river water was





pre-concentrated onto four 47 mm 0.2 μm polycarbonate filters using a gentle vacuum (less than 5 psi). Filters were transferred to 15 mL Falcon tubes containing 10 mL of river water and 500 μL of 10 % paraformaldehyde (PFA) for a final concentration of 0.5 % PFA. During preliminary investigations of Sacramento River water samples, we observed substantial loss of intact phytoplankton cells when freezing and thawing the pre-concentrated samples, and a degradation of Chl-*a*

florescence after >9 days of storage. Because both of these processes reduce the number of detectable phytoplankton cells in a sample over time and potentially impact $\delta^{15}N$ values of the sorted populations, all of the samples included in this study were sorted from unfrozen samples (stored in the dark at 4 °C) within a week of collection.

Phytoplankton were sorted with an Influx Cell Sorter in logarithmic mode (BD Biosciences, San Jose, CA, USA). Prior to

sorting, sample concentrates were pre-filtered using 50 μm mesh size filters to prevent clogging of the 70 μm diameter flow cytometer nozzle with large particles. Phytoplankton were detected and sorted into a gate using forward scatter (proxy for cell size) and chlorophyll fluorescence at 692 nm. Chlorophyll autofluorescence was excited using a 200 mW, 488 nm Sapphire laser (Coherent, Santa Clara, CA, USA). To ensure a sufficient mass of nitrogen, approximately 10 million cells were sorted directly into 5 mL polystyrene Falcon tubes orientated at a low angle to the sort stream. Regular analysis of sort

purity (calculated by analyzing a sorted sample to determine the proportion of events that fall within the target gate as a percentage of total event rate) was approximately 95 %. Because the river water contained abundant detritus and sediment, long sort times were required to achieve high sample purity. Unfortunately, the need to sort unfrozen samples within one week of collection precluded sorting sufficient cells for N for isotopic analysis of individual populations such as diatoms or cyanobacteria. Instead, all phytoplankton cells were sorted into a single sample from each location.

Sorted cells were transferred to 20 mL glass vials and dried down under vacuum using a centrifugal evaporator. Dried phytoplankton samples were redissolved in 20 μL ultra high purity deionized water and transferred into tin capsules. Capsules were dried overnight at 60 °C and then crushed into small cubes. $\delta^{15}N$ analysis of sorted phytoplankton was conducted using a Carlo Erba CHNS-O EA1108-Elemental Analyzer interfaced with a Thermo Finnigan Gasbench II

connected to an isotope ratio mass spectrometer (Thermo Fisher Scientific) at the UC Santa Cruz Stable Isotope Laboratory facility. The elemental analyzer and gas bench were configured to run small samples using a modification of the methods described in Polissar et al. (2009). In this configuration, samples as small as 35 nmol N could be analyzed with a precision of 0.5 ‰. Phytoplankton samples analyzed for this study ($\delta^{15}N$-PHY) ranged in size between 50 and 100 nmol N. Analysis of duplicate samples (sorted and analyzed independently) indicated a precision of 0.8 ‰ for the entire method.


### 2.5 Quantification of phytoplankton N source

A two end-member stable isotope mixing model approach was used to quantify changes in the N source used by phytoplankton encountering elevated $NH_4^+$ concentrations downstream of the WWTP. Assuming that $NO_3^-$ and $NH_4^+$ were





the only available N sources, the percentage of N uptake from $NH_4^+$ (%NH$_4$) was calculated according to Eq. 1, where $\delta^{15}N_{NO3}$, $\delta^{15}N_{NH4}$, $\delta^{15}N_{PHY}$ are the N isotopic ratios for $NO_3^-$, $NH_4^+$ and phytoplankton, and $\varepsilon_{NO3}$ and $\varepsilon_{NH4}$ are the enrichment factors for $NO_3^-$ and $NH_4^+$, respectively (York et al., 2007).

$$\%NH_4 = \frac{[(\delta^{15}N_{NO3}-\varepsilon_{NO3})-\delta^{15}N_{PHY}]}{[(\delta^{15}N_{NO3}-\varepsilon_{NO3})-(\delta^{15}N_{NH4}-\varepsilon_{NH4})]} \times 100 \tag{1}$$

The enrichment factor ($\varepsilon$) is an expression of the magnitude of fractionation between the substrate (e.g. $NH_4^+$ or $NO_3^-$) and product (e.g. phytoplankton). Phytoplankton are able to process $^{14}N$ faster than $^{15}N$ (due to a difference in energy required to

break bonds) and this results in lower $\delta^{15}N$ values in phytoplankton cells compared to their nutrient source as long as the nutrient source is not completely exhausted. In an open system when N substrates are not used to exhaustion, $\varepsilon$ can be approximated as the instantaneous difference between $\delta^{15}N_{substrate}$ and $\delta^{15}N_{product}$.

## 3 Results

During the October and June experiments, tidally-averaged flow in the Sacramento River was ~200 m$^3$ s$^{-1}$. Due to tidal
influence, river velocities at the top of the reach ranged from -0.06 m s$^{-1}$ to +0.40 m s$^{-1}$ while farther downstream river velocities ranged from -0.11 m s$^{-1}$ to +0.46 m s$^{-1}$, for an average velocity of about 0.18 m s$^{-1}$. Slack flow and flow reversals occurred around mid-day during the October sampling, and in the early morning and late afternoon during the June sampling. Water temperatures in October were ~16.5 ˚C, whereas the river was warmer in June (~22 ˚C). Chl-*a* concentration at the top of the study reach was 8 µg L$^{-1}$ in October and 20 µg L$^{-1}$ in June. During both sampling periods large
declines in Chl-*a* concentration were observed in all parcels as they travelled downstream starting ~16 km above the WWTP, such that by the time the parcels reached the most downstream sampling points Chl-*a* concentrations were about 2 µg L$^{-1}$ (Table 1), (Kraus et al., 2017).

### 3.1 Downstream trends in $NO_3^-$ and $NH_4^+$ concentrations

Nitrate concentrations increased downstream during the October and June campaigns in both +EFF and -EFF parcels, however, the downstream gains in $NO_3^-$ were more modest in -EFF parcels (Fig. 2). During both the October and June sampling campaigns, $NH_4^+$ concentrations were low (<1.0 µM) upstream of the WWTP. Immediately downstream of the WWTP $NH_4^+$ concentrations increased in the +EFF parcel to ~100 µM in October and ~60 µM in June. The maximum $NH_4^+$ concentration in October was higher than in June due to a greater percentage of effluent in the river (4.0 % in October
compared to 2.7 % in June) (Kraus et al., 2017). During the 3 days of downriver travel following effluent addition, riverine



$NH_4^+$ concentrations decreased modestly in the +EFF parcels. In contrast, $NH_4^+$ concentration remained less than 20 μM downstream of the WWTP in the -EFF parcels, and a slight increase in concentration was observed during downstream transit. In June, phytoplankton growth rates may have been N limited at the most upstream locations, as the concentration of DIN was close to the half-saturation constant of 7 μM frequently used to model phytoplankton growth (Travis et al., 2015).

## 3.2 Downstream trends in $\delta^{15}N$ of $NO_3^-$ and $NH_4^+$

Table 1 presents $\delta^{15}N$ values of $NO_3^-$ and $NH_4^+$ for all samples collected during the October and June campaigns with concentrations sufficient for analysis. During the October field campaign, $\delta^{15}N$-$NO_3$ started at 8.5 ‰ upstream of the WWTP and decreased downstream in both the +EFF and -EFF parcels. The magnitude of the decrease was greatest in the parcel

containing wastewater effluent; over 83 hours of travel downstream of the WWTP $\delta^{15}N$-$NO_3^-$ decreased by 7.5 ‰ in the +EFF parcel whereas $\delta^{15}N$-$NO_3$ only decreased by 1.6 ‰ in the -EFF parcel. In June, upstream values for $\delta^{15}N$-$NO_3^-$ were lower than in October (3-5 ‰), and remained relatively stable as both the +EFF and -EFF parcels travelled downstream.

Due to low concentrations of $NH_4^+$ upstream of the WWTP, it was only possible to measure $\delta^{15}N$-$NH_4^+$ in the +EFF parcels

downstream of the WWTP. In the +EFF parcels $\delta^{15}N$-$NH_4^+$ increased from 7.9 ‰ to 9.7 ‰ in October and from 8.0 ‰ to 10.7 ‰ in June with downstream travel. We also observed that $\delta^{15}N$-$NH_4^+$ increased while $NO_3^-$ concentration increased and $\delta^{15}N$-$NO_3$ values decreased during transit in parcels containing effluent, which suggests nitrification was occurring. This observation is consistent with high rates of nitrification previously reported in the Sacramento River (Hager and Schemel, 1992; Parker et al., 2012a; O'Donnell 2014; Damashek et al., 2016;).

## 3.3 Downstream trends in particulate organic matter and phytoplankton

$\delta^{15}N$-POM values decreased over the study reach in all parcels, though downstream trends were not monotonic: in particular, we observed periods during the night when $\delta^{15}N$-POM increased >2 ‰ (Fig. 3). The addition of effluent caused a larger decrease in $\delta^{15}N$-POM values during transit compared to the -EFF parcels in October and June. C:N atomic ratios in all POM

samples were near the Redfield ratio, ranging from 6.8 to 8.9, suggesting the POM was primarily phytoplankton (Tables 1 and 2). C:Chl-*a* (weight:weight) ratios of POM ranged from 10 to 170 for all samples, with a median value of 35, which is also consistent phytoplankton dominated POM during October and June (Table 1).

Similar to $\delta^{15}N$-POM, $\delta^{15}N$-PHY values decreased with downstream travel in all parcels; but the magnitude of decrease was

much greater in the +EFF parcels despite the fact that effluent contained $NH_4^+$ with a higher $\delta^{15}N$ value than $NO_3^-$ present upstream of the WWTP (Fig. 3). For many samples, the difference between $\delta^{15}N$-POM and $\delta^{15}N$-PHY was less than 1 ‰ (Fig. 4). However, $\delta^{15}N$-POM values diverged from $\delta^{15}N$-PHY after 20+ hours of travel past the WWTP in the +EFF parcels.




The largest difference between $\delta^{15}$N-POM and $\delta^{15}$N-PHY was ~7 ‰ in the October +EFF parcel at the most downstream site (Fig. 3, Fig. 4).

## 4 Discussion

### 4.1 Comparison of $\delta^{15}$N of POM and Phytoplankton

C:N and C:Chl-*a* ratios suggest that POM in the Sacramento River collected in October and June was primarily phytoplankton. Previous studies in the SFE have also reported POM C:N ratios near the Redfield Ratio (Canuel et al., 1995; Cloern et al., 2002), and in a survey of POM across the SFE, Wienke and Cloern (1987) reported a median C:Chl-a ratio of 50 over a range of phytoplankton community composition and productivity. The median C:Chl-*a* ratio of 35 observed in this study is also in agreement with previous studies which have used a C:Chl-a ratio of 35 as a conservative estimate of the

phytoplankton C to Chl-*a* ratio in the SFE (Cloern et al., 1995; Canuel et al., 1995; Sobczak et al 2005). Consistent with the interpretation that POM contained primarily phytoplankton, we found general agreement between $\delta^{15}$N-POM and $\delta^{15}$N-PHY, particularly in samples that did not contain effluent. However, we also observed a divergence between in $\delta^{15}$N-PHY and $\delta^{15}$N-POM values within 24 hours following the addition of effluent containing high concentrations of $NH_4^+$. The slower response to a change in N sources observed in bulk POM compared to intact phytoplankton isolated by flow cytometry

suggests that the bulk POM pool contained a significant fraction of dead or inactive phytoplankton not actively taking up nitrogen.

During the October and June experiments, phytoplankton samples were collected for quantitative enumeration and qualitative evaluation (for details see Kraus et al., 2017). During both experiments it was observed that diatoms accounted

for ~90% of the algal biovolume, and that upstream of the WWTP colonies appeared more vibrant compared to downstream samples which contained abundant decrepit cells and partially empty frustules (Kraus et al., 2017). A downstream decline in cell health, which mirrored decreasing Chl-*a* concentration, was observed in both +EFF and -EFF parcels. The observation of declining cell health in phytoplankton may help explain how bulk POM could contain primarily phytoplankton cells and yet also display a different downstream trend in $\delta^{15}$N values when compared to sorted phytoplankton. Because

phytoplankton cells were sorted from bulk POM based on a ratio of cell size and Chl-*a* florescence associated with healthy and intact cells, an increasing abundance of decrepit cells would not influence $\delta^{15}$N-PHY, but it could dilute the contribution of live phytoplankton to the $\delta^{15}$N value of bulk POM.

It is also possible that the presence of effluent caused an increase in heterotrophy by zooplankton and bacteria in the +EFF

parcels, resulting in $\delta^{15}$N-POM values greater than $\delta^{15}$N-PHY downstream of the WWTP. During the June experiment, zooplankton biomass was measured in the -EFF and +EFF parcels, and a decrease in zooplankton biomass was observed in the +EFF parcel downstream of the WWTP (Kraus et al., 2017). Thus, while we cannot rule out that possibility that



zooplankton growth elevated $\delta^{15}$N-POM downstream of the WWTP in October, it is unlikely that zooplankton caused the divergence between $\delta^{15}$N-POM and $\delta^{15}$N-PHY in the +EFF parcel in June. Heterotrophic bacterial abundance was not measured during this study, but temporal variability in $\delta^{15}$N-POM provides some indirect evidence of bacterial reworking of bulk POM during downstream transport. On multiple occasions in this study we observed that $\delta^{15}$N-POM increased

overnight in both +EFF and -EFF parcels. Isotopic fractionation during remineralization by bacteria has been shown to produce $NH_4^+$ ~3 ‰ lower than its organic matter source (Hoch et al, 1994), thus increasing the $\delta^{15}$N value of the remaining organic matter. If N remineralization exceeded uptake under low light conditions, that could explain observed increases in $\delta^{15}$N-POM at night. Remineralization of labile POM during downstream transport would also dilute the isotopic signal of $NH_4^+$ uptake by phytoplankton, resulting in $\delta^{15}$N-POM values greater than $\delta^{15}$N-PHY downstream of the WWTP in the

+EFF parcels. This interpretation would also be consistent with the results of a previous investigation into spatial and temporal variability of $\delta^{15}$N of POM in the freshwater portion of the SFE by Cloern et al. (2002). In that study, the authors reported little overlap between $\delta^{15}$N-POM values and the $\delta^{15}$N values of potential organic matter sources to POM, suggesting that bacterial processing had overprinted the isotopic composition of a significant fraction of the organic matter sources present in bulk POM.

### 4.2 Quantification of phytoplankton N source

To trace the movement of WWTP derived $NH_4^+$ into phytoplankton downstream of the WWTP, we employed a two-end member mixing model approach. The data used for mixing model calculations of phytoplankton N source using Eq. 1 are shown in Fig. 5. Use of this approach requires knowledge of enrichment factors for $NO_3^-$ ($\varepsilon_{NO3}$) and $NH_4^+$ ($\varepsilon_{NH4}$). Enrichment

factors for phytoplankton N use have been measured in numerous laboratory and field investigations (Cifuentes et al., 1989; Waser et al., 1998; Altabet et al., 1999; Needoba et al., 2003; Karsh et al., 2014). While a range of values have been reported for fractionation during assimilation, in general, $\varepsilon_{NO3}$ tends to be lower (2-7 ‰) than $\varepsilon_{NH4}$ (0-25 ‰). However, enrichment factors are known to be impacted by light, growth rate, and N concentration, and both N-limited conditions and high growth rates have been shown to result in lower enrichment factors (Finlay and Kendall, 2007).

There have been relatively few field investigations of $\varepsilon_{NO3}$ in fresh water settings. In single species culture studies, values of $\varepsilon_{NO3}$ from <1 ‰ to as high as 20 ‰ have been reported (Granger et al., 2004). In other coastal and estuarine field investigations $\varepsilon_{NO3}$ values between 2‰ and 7‰ have been reported (York et al., 2007 and references therein) and in a recent study in the Danube Delta a value of 2.7 ‰ was reported based on a Rayleigh distillation model (Möbius et al., 2015). In this

study, we estimated $\varepsilon_{NO3}$ from the difference between $\delta^{15}$N-$NO_3^-$ and $\delta^{15}$N-PHY in river water upstream of the WWTP where $NH_4^+$ concentrations are low and $NO_3^-$ is the dominant N source. The average offset between $\delta^{15}$N-$NO_3^-$ and $\delta^{15}$N-PHY in this portion of the river was 3 ‰ ($n = 6$), which fits within the range of previously reported values from culture and field investigations of $\varepsilon_{NO3}$ where growth was not nutrient limited. Field investigations of $\varepsilon_{NH4}$ are even less common than $NO_3^-$





investigations. Values ranging from 0 to 25 ‰ have been reported from both culture and field studies (Waser et al., 1999). In this study, the minimum value of $\varepsilon_{NH4}$ that resulted in solutions to Eq. 1 between 0 and 100% was 17 ‰ (using a $\varepsilon_{NO3}$ value of 3 ‰). An enrichment factor of 17‰ is relatively high compared to other studies, however previous investigations of $\varepsilon_{NH4}$ focused on nutrient limited conditions. Elevated $NH_4^+$ concentrations in this study may help explain the large apparent

enrichment factor.

The percentage of the phytoplankton N pool that was derived from $NH_4^+$ (%$NH_4$) versus $NO_3^-$ calculated using values of $\varepsilon_{NH4}$ ranging from 17 ‰ (the minimum value required for valid solutions to the model) to 25‰ (highest reported value) illustrate that model solutions become increasingly sensitive to $\varepsilon_{NH4}$ as $\delta^{15}$N-PHY decreases downstream. Nevertheless,

despite the uncertainty in enrichment factors, a gradual increase with downstream travel in the percentage of N derived from $NH_4^+$ is apparent (Fig. 6). In the October +EFF parcel the percentage of phytoplankton N derived from $NH_4^+$ increased to 50% over 60 hours of travel time and may have reached as high as 88% after 80 hours. Similar patterns were observed between the October and June samplings. However, mixing model calculations were only completed at three locations in June because $\delta^{15}$N-PHY was greater than $\delta^{15}$N-$NO_3^-$ and $\delta^{15}$N-$NH_4$ at the top of the study reach (discussed below).

Mixing model calculations suggest that downstream of the WWTP, a significant portion of the phytoplankton N pool was derived from $NO_3^-$ despite the presence of high concentrations of $NH_4^+$. However, N uptake experiments conducted using $^{15}$N-tracer incubation techniques as part of this effluent hold study indicate a near immediate switch from $NO_3^-$ uptake to $NH_4^+$ uptake when $NH_4^+$ concentrations were elevated in this portion of the river (Travis, 2015; Kraus et al., 2017). The

apparently gradual increase over time in the proportion of $\delta^{15}$N-PHY derived from $NH_4^+$ suggests either that simultaneous uptake of $NO_3^-$ and $NH_4^+$ occurs in the river under conditions not captured by the $^{15}$N-tracer incubations, or that the N turnover time of phytoplankton is much longer than the ~80 hour travel time covered in this study.

We can estimate potential N turnover time as the mean concentration of POM-N (µM) divided by the mean rate of N uptake

(µM d$^{-1}$) measured downstream of the WWTP during 24-h $^{15}$N-tracer experiments. This estimate assumes that all POM-N comes from phytoplankton and it represents a minimum turnover time because it is based on potential N uptake rates measured under high light conditions, in bottles where phytoplankton are isolated from the impacts of turbulence and mixing which may transport cells into lower light environments. In the October +EFF parcel, the concentration of particulate N (mean ± standard deviation) downstream of the WWTP was 4.0 ± 1.0 µM, while the potential $NH_4^+$ uptake rate was 1.4 ± 0.6

µM d$^{-1}$ (mean ± standard deviation) (Travis, 2015). This implies that it would take ~66 hours, to completely turnover phytoplankton N with newly assimilated $NH_4^+$ if phytoplankton switched to 100 % $NH_4^+$ uptake. After 60 hours of travel time in the presence of elevated of $NH_4^+$ concentration, mixing model calculations indicate that only ~50 % of the phytoplankton N was derived from $NH_4^+$, suggesting $NH_4^+$ uptake rates in the river are much lower than the potential growth rates measured in $^{15}$N uptake experiments. Given that phytoplankton in the river likely experienced light limitation, it is



reasonable to infer that lower *in situ* growth rates resulted in an N turnover time greater than 80 hours. Because $\delta^{15}$N-PHY reflects a time integrated mixture of N uptake, an N turnover time >80 hours would mute changes in $\delta^{15}$N-PHY following an abrupt switch to $NH_4^+$ uptake.

While there is a general consensus that phytoplankton preferentially take up $NH_4^+$ when $NH_4^+$ concentrations are elevated (for reviews see Dortch, 1990 and; Glibert et al., 2016), simultaneous uptake of $NO_3^-$ and $NH_4^+$ has been documented in several field studies. For example, Berg et al. (2001) report nearly equal percentages of $NO_3^-$ and $NH_4^+$ uptake with a small percentage of N uptake as urea for the spring bloom diatom *Thalassiosira baltica*. Likewise Twomey et al. (2005) report near parity of uptake of N as $NO_3^-$ and $NH_4^+$ for the phytoplankton community in the Neuse River Estuary. Both field and

laboratory based studies make it clear that different cells respond differently to the presence of multiple N sources (Dortch, 1990). Diatoms, for example, can reach maximum growth rates when using both $NO_3^-$ and $NH_4^+$, whereas cyanobacteria appear to be $NH_4^+$ specialists (Senn and Novick, 2014, and references therein). During this study, diatoms were the most abundant type of algae, which would be consistent with simultaneous use of $NO_3^-$ and $NH_4^+$. However, $^{15}$N-tracer uptake measurements made during 24-h bottle incubations in this section of the Sacramento River have consistently found that $^{15}$N-

$NO_3$ uptake rates are near zero when $NH_4^+$ concentrations are elevated (Parker et al., 2012a; Kraus et al., 2017) meaning that if simultaneous uptake of $NO_3^-$ and $NH_4^+$ occurred in the river it would appear to require conditions (such as light limitation) not captured in these incubations.

Another possible explanation for the observed gradual increase in the $\%NH_4$ making up the phytoplankton N pool in the

+EFF parcels is that $\delta^{15}$N-PHY represents a mixture of phytoplankton actively taking up $NH_4^+$ (as observed in bottle incubations) and phytoplankton subsisting on an internal supply of $NO_3^-$ acquired upstream of the WWTP. Previous investigations have shown that diatoms are capable of accumulating an internal DIN pool under both N-sufficient and N-deficient conditions and that DIN accumulation is impacted by prior conditioning of the cells (Dortch, 1982; Collos, 1982; Lomas and Glibert, 2000). During the June transect $\delta^{15}$N-PHY was greater than $\delta^{15}$N-$NO_3$ at the top of the study reach and

remained greater than both $\delta^{15}$N-$NO_3$ and $\delta^{15}$N-$NH_4$ for >40 hours of downstream transport. Because $\delta^{15}$N-PHY should be equal to or lower than the $\delta^{15}$N value of its source N, it appears that during the June experiment phytoplankton N was acquired above the study reach. A similar observation was made in the Childs River of Massachusetts, where phytoplankton maintained a stable $\delta^{15}$N value over several days of downstream transport while both $\delta^{15}$N-$NO_3$ and $\delta^{15}$N-$NH_4$ decreased, leading York et al. (2007) to infer that phytoplankton growth was sustained by internal N stores. If a portion of the

phytoplankton community acquired N upstream of the study reach and then was advected downstream without taking up additional N this could account for the 20% apparent contribution of $NO_3^-$ to $\delta^{15}$N-PHY after 80 hours of transport.





## 5 Conclusions

Monitoring the spatial influence and biological uptake of anthropogenic nutrient loading in aquatic ecosystems is a pressing resource management challenge. Natural abundance stable isotope approaches have the potential to help regional monitoring programs with this challenge if applied in well-characterized systems. In this study, we took advantage of a river-scale
nutrient manipulation experiment to test the use of flow cytometry to isolate phytoplankton from bulk POM prior to isotopic analysis. Comparison of $\delta^{15}$N-POM and $\delta^{15}$N-PHY revealed that POM and phytoplankton share similar downstream trends in the Sacramento River, suggesting that POM (which is relatively easy to collect and analyze) may be a useful proxy for phytoplankton under certain conditions. However, where phytoplankton growth rates are low, or N sources change abruptly, $\delta^{15}$N-POM may not reflect localized changes in $\delta^{15}$N-PHY, which could lead to inaccurate interpretation of the relative
importance of different N sources if not carefully considered.

Isolating phytoplankton allowed use of a mixing model approach to trace the movement of WWTP $NH_4^+$ into the phytoplankton N pool. We found that even in the presence of high concentrations of $NH_4^+$, where $^{15}$N uptake experiments suggest preferential uptake of $NH_4^+$ and little to no $NO_3^-$ uptake, a large portion of phytoplankton N (10-60 %) was derived
from $NO_3^-$ following several days of downstream transport (Fig. 6). Differences observed between the natural abundance and $^{15}$N-labeled approaches highlight the strengths and weaknesses of both methods. The strength of the natural abundance approach is that it allows *in situ* observation, thus avoiding the potential artifacts associated with altered conditions such as increased light availability and a lack of turbulence and grazing that may impact phytoplankton populations in bottle incubations. A significant drawback of the natural abundance approach is that it integrates all N use up to the point of
sampling thus potentially complicating interpretation of short term (~24 h) changes in N sources. When the results of both approaches are considered together, it appears that *in situ* growth rates were much lower in the river than observed in bottle incubations, leading to a slow turnover of phytoplankton N and a gradual change in the $\delta^{15}$N-PHY.

Results of this study indicate that flow cytometry coupled with natural abundance stable isotope techniques can provide
valuable insight into how different nutrient sources enter the food web. Obtaining pure phytoplankton samples in the presence and absence of effluent allowed us to determine that the presence of WWTP effluent containing $NH_4^+$ with a distinctly high $\delta^{15}$N value resulted in a decrease in $\delta^{15}$N-PHY values due to large enrichment factors in this nutrient replete setting. One implication of this finding is that planned upgrades to the Sacramento River WWTP (including nitrification and denitrification), which will reduce $NH_4^+$ inputs to the SFE by 2021, may actually result in an increase in $\delta^{15}$N-PHY. This
increase may subsequently be transferred up the food chain. Results from this study provide an important baseline for future stable isotope investigations of nutrient flow in the SFE following WWTP upgrades.

While flow cytomtery allowed for determination of $\delta^{15}$N-PHY separate from bulk POM, we did encounter challenges in applying this approach in a riverine setting which warrant further exploration and method development. Unfortunately, due to abundant sediment as well as fragile phytoplankton cells, we were not able to complete isotopic analysis of distinct phytoplankton populations within bulk POM. The limitation was not sample size, but rather the time required to sort

sufficient material before the sample degraded. Future investigations could avoid this issue by combining sorted samples collected over several days or weeks. This approach would reduce temporal resolution, but this reduction may actually be appropriate given that low phytoplankton growth rates observed in this study complicated interpretation of changes in $\delta^{15}$N-PHY over shorter timescales.

Additional research is needed to establish sampling strategies that allow for sorting of different populations of phytoplankton (such as diatoms) or bacteria from bulk POM. For example, isotopic data collected in this study, particularly downstream trends in $\delta^{15}$N-NH$_4$ and $\delta^{15}$N-NO$_3$ as well as the daily temporal variations in $\delta^{15}$N-POM, suggest that additional N cycling processes such as nitrification and remineralization influence N source availability for phytoplankton. If future studies focused on sorting unique populations of bacteria and phytoplankton, it would be possible to isotopically trace the pathways

by which N becomes available to phytoplankton. This could greatly improve our understanding of natural and anthropogenic cycling of N in aquatic systems.

**Author Contributions**

C. Schmidt wrote proposals, participated in field sample collection, and was responsible for flow cytometry and stable

isotope analyses at UCSC. T. Kraus wrote proposals, designed and managed field experiments, led data compilation and participated in data interpretation. M. Young completed stable isotope analysis at the USGS and participated in data interpretation. C. Kendall wrote proposals and participated in data interpretation.  C. Schmidt prepared the manuscript with contributions from all co-authors.

**Competing Interests**

The authors declare that they have no conflict of interest.

**Acknowledgements**

Special thanks to numerous staff at the USGS California Water Science Center for orchestrating and conducting the Lagrangian sampling campaigns, particularly Brian Bergamaschi, Bryan Downing, Elizabeth Stumpner, and Katy

O'Donnell. Also special thanks to Alex Parker, Nicole Stern and Frances Wilkerson and the RTC lab staff who analyzed samples for nutrients, chlorophyll, and C and N uptake.  Thank you to Kurt Carpenter and Joe Fackrell for providing useful



comments on an earlier draft. We are also grateful for the help of Dyke Andreason and Brandon Carter for their analytical support at UC Santa Cruz.

Support for this work was provided by Sacramento County Regional Sanitation District (Grant Number: 13WSCA600000947/90000080), the Interagency Ecological Program for the San Francisco Estuary (Grant Number: 13WSCA4600010038/4600010038), and California Sea Grant (Grant Number: DelSci U-04-SC-005)

Any use of trade, firm, or product names is for descriptive purposes only and does not imply endorsement by the U.S. Geological Survey.

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



**Table 1.** Summary of conditions in the two water parcels sampled in October 2013; one parcel did not receive effluent (-EFF) and one parcel did receive effluent (+EFF) as it passed the wastewater treatment plant (WWTP). A travel time of zero represents the time each parcel passed the location where effluent from the WWTP enters the river.

| Parcel | Travel Time (h) | Chl-a (μg/L) | [NO$_3^-$] (μM) | [NH$_4^+$] (μM) | δ$^{15}$N-NO$_3$ (‰) | δ$^{15}$N-NH$_4$ (‰) | POM C:N (atomic) | C:Chl-a (wt:wt) | δ$^{15}$N-POM (‰) | δ$^{15}$N-PHY (‰) |
|---|---|---|---|---|---|---|---|---|---|---|
| -EFF | -18.9 | 8.0 | 2.8 | 0.6 | | | 6.9 | 24.2 | 6.3 | |
| -EFF | -14.2 | 5.4 | 2.7 | 0.7 | | | 6.4 | 27.9 | 4.9 | |
| -EFF | -12.0 | 7.1 | 2.8 | 0.8 | 8.8 | | 6.8 | 22.5 | 6.0 | 6.5 |
| -EFF | 5.0 | 4.0 | 3.2 | 1.4 | 9.1 | | 6.8 | 33.7 | 6.4 | 6.2 |
| -EFF | 8.2 | 3.3 | 3.3 | 1.2 | | | 6.2 | 36.6 | 3.0 | |
| -EFF | 10.3 | 2.9 | 3.1 | 0.8 | | | 7.2 | 56.8 | 5.3 | |
| -EFF | 12.9 | 4.9 | 3.4 | 1.9 | 8.6 | | 7.4 | 35.3 | 4.8 | 6.6 |
| -EFF | 29.5 | 3.5 | 4.4 | 4.3 | 8.1 | | | 20.5 | | 4.6 |
| -EFF | 32.5 | 3.5 | 3.5 | 1.2 | | | 7.4 | 40.4 | 5.6 | |
| -EFF | 34.5 | 3.0 | 5.4 | 7.8 | | | 6.6 | 26.4 | 2.3 | |
| -EFF | 36.9 | 5.4 | 7.7 | 7.0 | | | 7.7 | 36.6 | 2.3 | |
| -EFF | 53.6 | 4.3 | 6.4 | 2.1 | 7.6 | | 7.7 | 21.3 | 4.6 | 4.4 |
| -EFF | 56.0 | 4.0 | 8.7 | 2.4 | | | 7.6 | 32.5 | 3.5 | |
| -EFF | 59.0 | 6.4 | 11.1 | 18.2 | | | 7.1 | 30.3 | -0.5 | |
| -EFF | 61.0 | 6.1 | 15.3 | 24.4 | | | 7.7 | 19.3 | 0.3 | |
| -EFF | 77.1 | 1.8 | 14.0 | 5.2 | | | 7.9 | 23.2 | -2.5 | |
| -EFF | 80.2 | 1.6 | 9.4 | 4.9 | | | 7.8 | 72.3 | 2.9 | |
| -EFF | 82.5 | 2.1 | 9.8 | 3.0 | | | 7.9 | 28.9 | 2.2 | |
| -EFF | 85.5 | 3.3 | 9.0 | 6.1 | 7.1 | | 7.9 | 38.6 | 2.2 | 2.8 |
| | | | | | | | | | | |
| +EFF | -18.7 | 4.7 | 3.6 | 0.4 | | | 6.8 | 20.0 | 7.8 | |
| +EFF | -14.5 | 7.6 | 3.6 | 0.5 | | | 7.3 | 19.7 | 6.8 | |
| +EFF | -12.5 | 6.3 | 3.5 | 0.5 | 9.3 | | 7.1 | 21.0 | 7.4 | 7.8 |
| +EFF | 3.3 | 4.3 | 4.0 | 98.5 | 6.7 | 7.9 | 7.2 | 57.4 | 3.2 | 5.3 |
| +EFF | 6.3 | 2.8 | 4.4 | 101.3 | | | 6.9 | 71.2 | 1.3 | |
| +EFF | 9.1 | 7.3 | 4.7 | 92.0 | 6.5 | 8.3 | 6.8 | 27.9 | 1.2 | 1.8 |
| +EFF | 11.7 | 4.2 | 4.9 | 94.5 | | | 6.4 | 37.5 | 1.2 | |
| +EFF | 29.2 | 2.1 | 6.7 | 63.3 | 5.4 | 8.8 | 6.8 | 65.7 | 1.2 | |
| +EFF | 32.0 | 3.1 | 7.0 | 77.0 | | | 7.2 | 53.5 | -1.0 | |
| +EFF | 34.0 | 2.0 | 7.5 | 76.5 | 4.2 | 8.3 | 7.1 | 70.0 | -0.2 | -2.0 |
| +EFF | 52.3 | 2.1 | 13.0 | 86.3 | | | 7.5 | 48.7 | 1.7 | |
| +EFF | 55.9 | 4.3 | 13.8 | 82.7 | | | 8.3 | 26.2 | -0.4 | |
| +EFF | 59.0 | 2.3 | 15.6 | 85.1 | 2.6 | 9.4 | 8.5 | 49.5 | -1.0 | -3.5 |
| +EFF | 76.7 | 2.5 | 28.8 | 71.2 | 1.7 | 9.9 | 8.0 | 38.1 | 0.5 | -6.4 |
| +EFF | 78.9 | 1.6 | 9.4 | 77.2 | | | 9.8 | 53.4 | -0.5 | |
| +EFF | 82.6 | 1.1 | 20.0 | 78.0 | 1.8 | 9.7 | 8.2 | 87.4 | -0.1 | |



**Table 2. Summary of conditions in the two parcels sampled in June 2014; one parcel did not receive effluent (-EFF) and one parcel did receive effluent (+EFF) as it passed the wastewater treatment plant (WWTP). A travel time of zero represents the time each parcel passed the location where effluent from the WWTP enters the river.**

| Parcel | Travel Time (h) | Chl-a (µg/L) | $[NO_3^-]$ µM | $[NH_4^+]$ µM | $\delta^{15}N$-$NO_3$ (‰) | $\delta^{15}N$-$NH_4$ (‰) | POM C:N (atomic) | C:Chl-$a$ (wt:wt) | $\delta^{15}N$-POM (‰) | $\delta^{15}N$-PHY (‰) |
|---|---|---|---|---|---|---|---|---|---|---|
| -EFF | -26.1 | 20.7 | 0.4 | 0.6 | 3.8 | | 7.0 | 10.3 | 5.7 | 2.1 |
| -EFF | -23.4 | 10.5 | 0.4 | 0.3 | | | 7.1 | 35.4 | 6.2 | |
| -EFF | -20.6 | 8.8 | 0.8 | 0.8 | 3.5 | | 7.6 | 41.5 | 5.1 | 4.7 |
| -EFF | -2.4 | 6.6 | 1.5 | 1.7 | | | 6.8 | 19.5 | 6.1 | 4.6 |
| -EFF | 0.7 | | 1.8 | 1.4 | | | 7.1 | | 5.7 | |
| -EFF | 3.3 | 4.7 | 3.3 | 1.6 | 4.3 | | 8.4 | 33.9 | 3.9 | 5.5 |
| -EFF | 21.9 | 3.8 | 2.3 | 2.0 | 4.9 | | 7.4 | 26.3 | 4.1 | 6.0 |
| -EFF | 24.8 | | 2.1 | 1.9 | 4.3 | | 7.9 | | 4.0 | |
| -EFF | 27.9 | 4.3 | 1.5 | 1.8 | 5.1 | | 8.0 | 68.7 | | 4.0 |
| -EFF | 45.8 | 2.6 | 3.2 | 11.0 | 3.6 | | 7.2 | 36.8 | 0.4 | 3.5 |
| -EFF | 49.4 | | 3.0 | 3.7 | 4.4 | | 8.9 | | 0.3 | 3.3 |
| | | | | | | | | | | |
| +EFF | -25.1 | 15.9 | 1.1 | 1.0 | 3.9 | | 7.2 | 10.5 | 6.5 | 7.4 |
| +EFF | -22.7 | 13.2 | 0.6 | 1.0 | | | 7.1 | 14.0 | 7.6 | |
| +EFF | -19.7 | 11.6 | 0.5 | 0.6 | | | 7.3 | 32.8 | 7.2 | 6.4 |
| +EFF | -1.5 | 4.9 | 1.3 | 1.4 | | | 7.3 | 34.5 | 3.9 | 5.1 |
| +EFF | 2.3 | 7.6 | 2.3 | 55.1 | | 8.0 | 7.3 | 59.8 | 1.1 | |
| +EFF | 4.3 | 5.3 | 2.6 | 58.8 | 3.4 | 8.3 | 7.5 | 100.1 | 1.3 | |
| +EFF | 23.5 | 4.3 | 5.3 | 53.0 | 3.2 | 9.0 | 7.4 | 34.1 | 1.5 | -4.0 |
| +EFF | 26.2 | | 6.3 | 44.1 | | 8.8 | 7.9 | 161.8 | 1.0 | |
| +EFF | 28.2 | 2.9 | 7.4 | 48.6 | 3.6 | 8.7 | 8.1 | 115.8 | 0.4 | -4.8 |
| +EFF | 51.0 | | 10.8 | 43.4 | 3.1 | 10.7 | 8.8 | 174.6 | 1.4 | -3.7 |




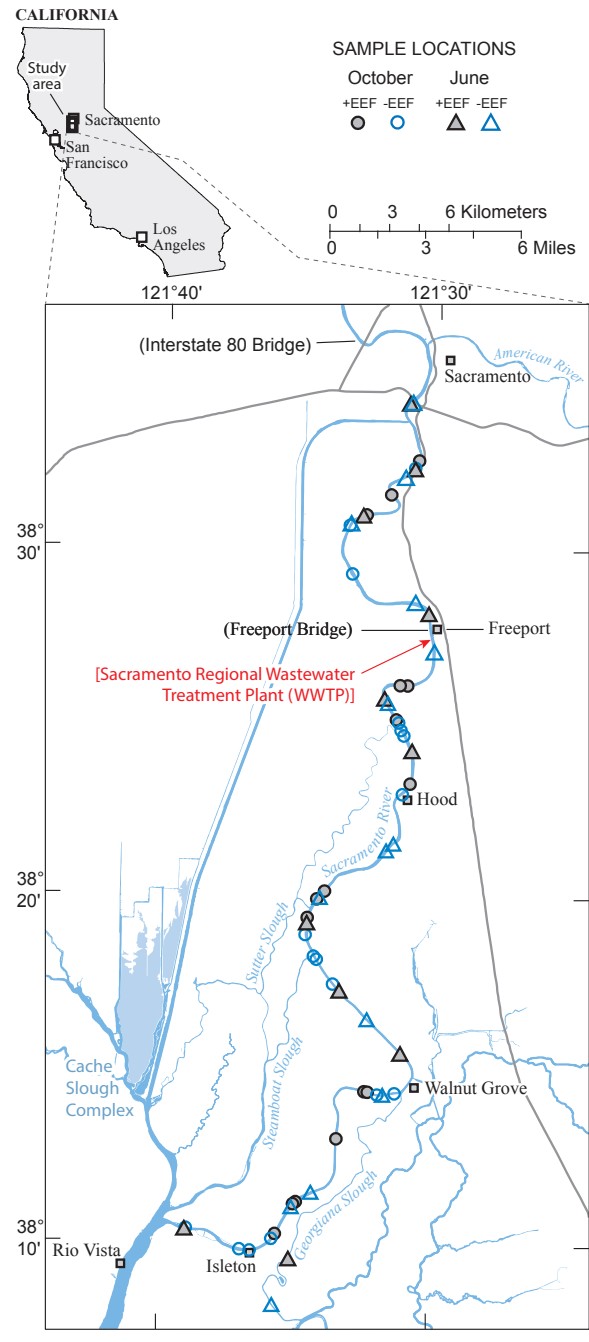

**Figure 1. Map of the study reach on the lower 70 km of the Sacramento River, CA showing the location where effluent from the Sacramento Regional Wastewater Treatment Plant (WWTP) enters the river and the locations of samples collected during the October and June experiments.**




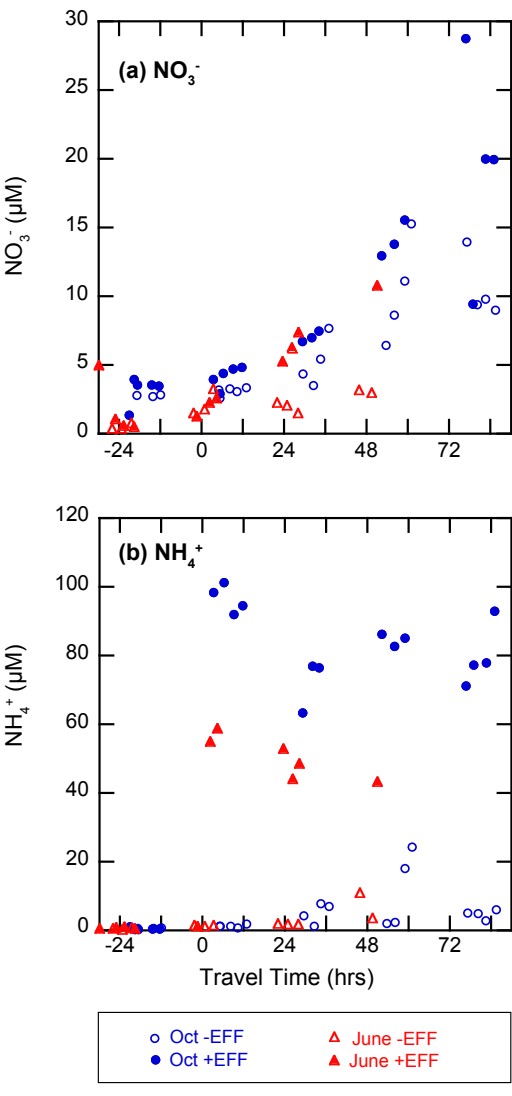

**Figure 2. Concentration of NO$_3^-$ (a) and NH$_4^+$ (b) in samples collected during the October and June Lagrangian experiments from parcels that either received effluent (+EFF, filled symbols) or did not receive effluent (-EFF, open symbols) as they traveled past the WWTP. Samples are plotted by travel time, where zero represents the time the parcel passed the location where effluent high in NH$_4^+$ enters the river.**



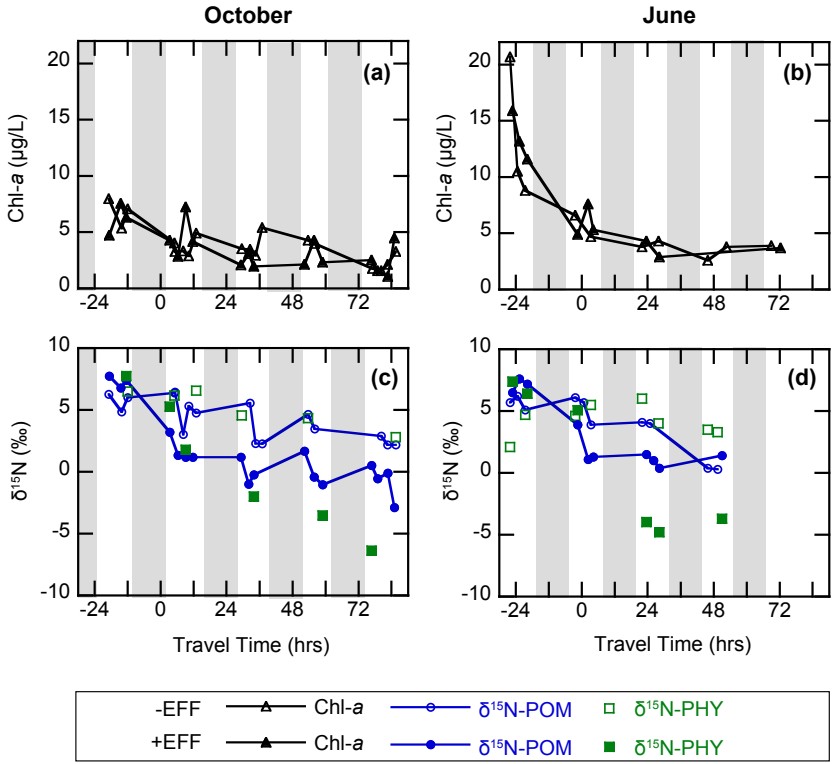

**Figure 3. Comparison of Chl-*a*, δ¹⁵N-POM, δ¹⁵N-PHY in parcels sampled in June (a,c) and October (b, d) for parcels that did receive effluent from the WWTP (+EFF) and parcels that did not receive effluent (-EFF). Samples are plotted by travel time, where zero indicates the time the parcel passed the location where effluent high in $NH_4^+$ enters the river. Shaded areas indicate nighttime.**



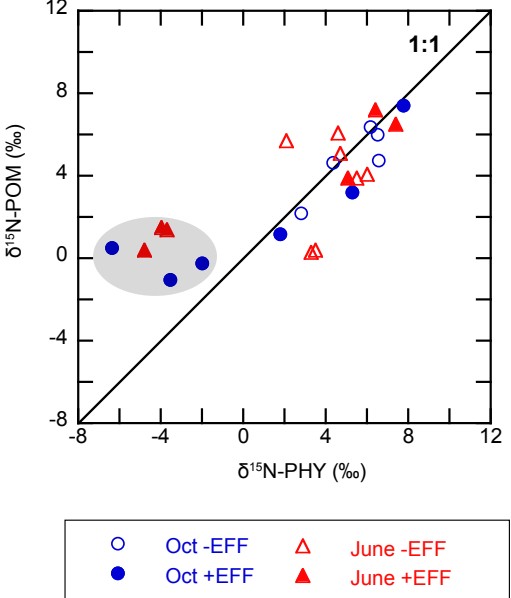

**Figure 4. Comparison of δ¹⁵N-POM and δ¹⁵N-PHY for October and June experiments for parcels that received effluent (+EFF) and parcels that did not receive effluent (-EFF). The grey circle indicates the six +EFF samples collected >20 hours downstream of the WWTP.**



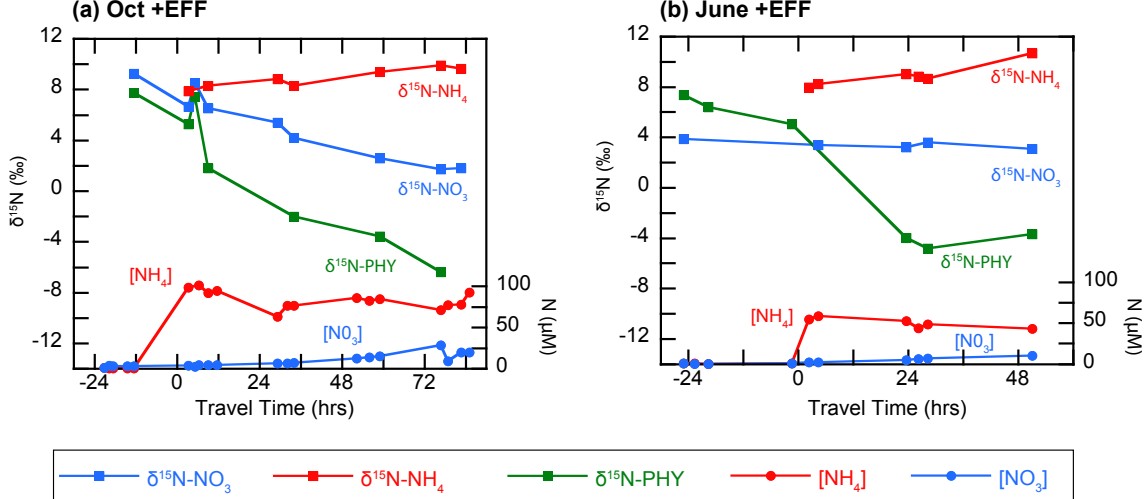

**Figure 5. $\delta^{15}$N-NO$_3$, $\delta^{15}$N-NH$_4$, and $\delta^{15}$N-PHY in parcels that received effluent from the WWTP (+EFF) in October (a) and June (b). Samples are plotted by travel time, where zero indicates the time the parcel passed the location where effluent high in NH$_4^+$ enters the river. The concentration of NH$_4^+$ was too low to allow for isotopic analysis upstream of the WWTP.**




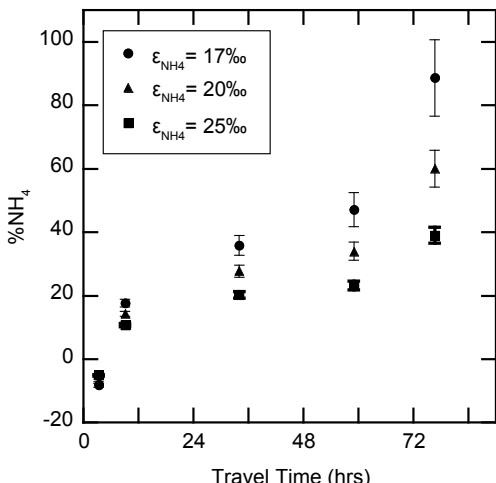

**Figure 6. Modeled percentage of phytoplankton N sourced form NH$_4^+$ (%NH$_4$) versus travel time past the wastewater treatment plant in the October parcel that received effluent (+EFF). Calculations were made using an enrichment factor for NO$_3^-$ of 3 ‰, and three different enrichment factors for NH$_4^+$ (17 ‰, 20 ‰, and 25 ‰). Error bars indicate propagated error from the ± 0.8 ‰ uncertainty in δ$^{15}$N-PHY values. See text for details.**