# Peer review of "Use of flow cytometry and stable isotope analysis to determine phytoplankton uptake of wastewater derived ammonium in a nutrient-rich river"

_Biogeosciences, 2017_

## Referee Comment (RC1) · Anonymous Referee #1 · 8 Aug 2017

General comment :

This study aims to trace the dissolved inorganic nitrogen source primarily used by phytoplankton in a river impacted by an heavy anthropogenic nutrient (ammonium) enrichment. The authors report an interesting dataset of stable nitrogen isotope ratio measurement in several inorganic and organic nitrogen pool. Moreover, they used a novel and elegant method (combination of flow cytometry cell sorting with stable isotope analysis) in order to distinguish (healthy) phytoplankton cells from the bulk particulate organic matter. The manuscript is well-written and the results reported by the

authors are appropriately discussed, overall I greatly enjoyed reading this manuscript.

Specific comment :

*p4, line 26 : No results from a 15N-labeled nutrient uptake experiment are described in this manuscript, hence I would suggest to remove this from the material & method section.

*p5, line 1-5 : Please, provide more information about the methodologies used to measure $NO_2^-$, $NO_3^-$ and $NH_4^+$ concentration ( the chemistry behind).

*p5, line 6 : I don't understand to what "70$\mu$m" is related. As you certainly know, nominal pore size of of the GFF filters is 0.7 $\mu$m. typo ?

*p8, line 6-19 : I would suggest to plot the d15N-$NO_3^-$ and d15N-$NH_4^+$ data in a way similar to figure 3 (ie. Data plotted against travel time). These data are interesting, but it is difficult to visualize the trend when looking at table 1 only.

*p9, line 25 : typo : fluorescence, and not florescence.

*p10, line 2 : Higher importance of labile POM mineralization in the +EFF parcel seems indeed plausible. It is a bit unfortunate that you did not measure heterotrophic bacteria abundance, but did you measure dissolved oxygen concentration, or any other variable related to ecosystem metabolism (for instance, community respiration) ? They might be helpful to directly put in evidence a putative higher importance of heterotrophic metabolism in the +EFF parcels.

*p10, line 16 and below : I understood reading your paper that the diatom "health status" was decreasing downstream, then could it be hypothesized that a change in the composition of the phytoplankton assemblage downstream of the location where the effluent enter the Sacramento river explains the gradual increase in the contribution of $NH_4^+$ ? Did you look at the phytoplankton composition (and assess its variability) at several location during the travel of the two parcels of water downstream ?

*p10, line 16 and below : Beside NH4+ and NO3-, N2 fixation could also be a significant N source in systems where cyanobacteria are abundant. Do you know what was the contribution of cyanobacteria to the phytoplankton assemblage ? Could you explain why you rule out any contribution of N2 fixation as a N source in the Sacramento river ?

---

## Referee Comment (RC2) · Anonymous Referee #2 · 19 Aug 2017

The manuscript by Schmidt et al. provides an interesting case study of the application of a very promising and little-used technique, by using flow cytometry as a sample preparation step to analyse d15N of phytoplankton to study nitrogen cycling in river systems . The ms is well written and data interpretation is generally solid.

I have relatively minor suggestions for improvement:

-While the focus of the manuscript is evidently on nitrogen cycling, it might be interesting to also report and discuss d13C values for the flow cytometry sorted phytoplankton

samples – I assume these were analyses in the same run ?

-the authors sampled following a Lagrangian approach – but it should perhaps be mentioned somewhere that the residence time of particles in a river system is expected to be higher than the water travel time in a river system, and some discussion on how that might affect the interpretation of results.

-page 5, line 7: I assume this should be 0.7 $\mu$m GFF filters (not 70 $\mu$m) ?

-Methods: while the authors refer to Polissar et al. (2008) for the 'micro-EA' setup, the latter does not use a GasBench as interface, and the authors provide no details on the trapping/focussing of the eluting gases. Some more details would be of interest to readers who wish to setup a similar configuration.

-The nutrient concentration profiles clearly suggest that nitrification could be an important process affecting nutrient cycling in this system – it would be worth discussing this aspect and checking if there are data that might shed some light on this. Are there dissolved oxygen data available ? Have others measured nitrification rates in this system ?

---

## Author Comment (AC1) · 14 Sep 2017

General comment : This study aims to trace the dissolved inorganic nitrogen source primarily used by phytoplankton in a river impacted by an heavy anthropogenic nutrient (ammonium) enrichment. The authors report an interesting dataset of stable nitrogen isotope ratio measurement in several inorganic and organic nitrogen pool. Moreover, they used a novel and elegant method (combination of flow cytometry cell sorting with stable isotope analysis) in order to distinguish (healthy) phytoplankton cells from the bulk particulate organic matter. The manuscript is well-written and the results

reported by the authors are appropriately discussed, overall I greatly enjoyed reading this manuscript.

Specific comment:

*p4, line 26 : No results from a 15N-labeled nutrient uptake experiment are described in this manuscript, hence I would suggest to remove this from the material & method section.

AC: This text has been removed.

*p5, line 1-5 : Please, provide more information about the methodologies used to measure NO2-, NO3- and NH4+ concentration (the chemistry behind).

AC: The following information will be added to the methods section:

"Nitrite concentration was determined by the Greiss reaction in which sulfanilamide and N-(1-Naphthy1) ethylenediamine dihydrochloride (NNED) reacts with nitrite in aqueous acidic solution to form an intensely pink diazo dye with an absorption maximum at 540 nm (Bendschneider and Robinson, 1952). To determine nitrate concentration, nitrate was reduced to nitrite by passage through a column containing copperized cadmium filings (Wood, Armstrong and Richards, 1967), and then the concentration of nitrate plus nitrite was determined in the manner described above. NH4+ concentrations were measured spectrophotometrically using a 10-cm path length cell according to the phenolhypochlorite method described in Solorzano (1969)."

*p5, line 6 : I don't understand to what "70_m" is related. As you certainly know, nominal pore size of of the GFF filters is 0.7 _m. typo ?

AC: Typo corrected, GFF filters had a nominal 0.7 $\mu$m pore size.

*p8, line 6-19 : I would suggest to plot the d15N-NO3- and d15N-NH4+ data in a way similar to figure 3 (ie. Data plotted against travel time). These data are interesting, but it is difficult to visualize the trend when looking at table 1 only.

AC δ15N-NO3- and δ15N-NH4+ are in fact plotted against travel time in Figure 5. A reference to figure 5 will be added to the paragraph to improve clarity.

*p9, line 25 : typo : fluorescence, and not florescence.

AC: Typo corrected.

*p10, line 2 : Higher importance of labile POM mineralization in the +EFF parcel seems indeed plausible. It is a bit unfortunate that you did not measure heterotrophic bacteria abundance, but did you measure dissolved oxygen concentration, or any other variable related to ecosystem metabolism (for instance, community respiration) ? They might be helpful to directly put in evidence a putative higher importance of heterotrophic metabolism in the +EFF parcels.

AC: Thank you for the helpful suggestion. We did measure dissolved oxygen (DO) concentration during the study, which reveals that DO was consistently lower in the parcels containing effluent (see figure S3D from Kraus et al. 2017, included below), thus providing additional support for the argument that heterotrophic metabolism differed between +EFF and –EFF parcels.

This point will be clarified in the manuscript:

"It is also possible that the presence of effluent caused an increase in heterotrophy by zooplankton and bacteria in the +EFF parcels, resulting in δ15N-POM values greater than δ15N-PHY downstream of the WWTP. Heterotrophic bacterial abundance was not measured during this study, but dissolved oxygen concentration was monitored, and in both October and June experiments dissolved oxygen concentrations were lower in the parcels containing effluent (Kraus et al., 2017)."

*p10, line 16 and below : I understood reading your paper that the diatom "health status" was decreasing downstream, then could it be hypothesized that a change in the composition of the phytoplankton assemblage downstream of the location where the effluent enter the Sacramento river explains the gradual increase in the contribution

of NH4+ ? Did you look at the phytoplankton composition (and assess its variability) at several location during the travel of the two parcels of water downstream ?

AC: This isotopic study was conducted in parallel with a larger investigation into this very important question and the results are reported in Kraus et al. 2017. To summarize: Whole-water samples for phytoplankton enumeration were collected at each sampling point during downstream travel (15 points in October, 9 points in June). Patterns in the phytoplankton assemblages were examined using nonmetric multidimensional scaling (NMDS) ordinations constructed from Bray–Curtis similarity matrices (square root- transformed abundance data). Potential differences between the +EFF and -EFF parcels were tested for significance using analysis of similarity (ANOSIM). The results of this analysis showed a significant separation of phytoplankton assemblages between October and June, but no statistical difference between +EFF and –EFF parcels in October or June.

Given this finding we concluded it was unlikely that changes in community composition cause the greater increase in %NH4 in +EFF parcels compared to –EFF parcels. To clarify this point, we will add the following sentence to the discussion section on pg 9 line 19:

"Patterns in the phytoplankton assemblages were examined and potential differences between the +EFF and -EFF parcels were tested for significance using analysis of similarity (ANOSIM). The results of this analysis showed no statistical difference between the assemblages present in +EFF and –EFF parcels in October or June (Kraus et al. 2017)."

*p10, line 16 and below : Beside NH4+ and NO3-, N2 fixation could also be a significant N source in systems where cyanobacteria are abundant. Do you know what was the contribution of cyanobacteria to the phytoplankton assemblage ? Could you explain why you rule out any contribution of N2 fixation as a N source in the Sacramento river?

AC: We are not aware of any studies that have looked at N2 fixation in the Sacramento

River. However, it is generally believed that N2 fixation is not a significant process in large rivers like the Sacramento River, which is a deep (4-10 m), swift moving, channelized river lined with rip rap, which also has high concentrations of DIN. We consulted with several colleagues on this topic. For example Vitousek et al. 2002 states "The few studies of nitrogen fixation in flowing water systems suggest that cyanobacteria and N fixation are limited by some of the same factors limiting other lotic algae: low light in small, heavily shaded streams (Horne & Carmiggelt 1975) and high current velocity (small to mid-sized streams) or high turbidity (large rivers). Indeed, planktonic cyanobacteria do not occur at all in most streams."

Vitousek, P. M., Cassman, K., Cleveland, C., Crews, T., Field, C. B., Grimm, N. B., Howarth, R. W., Marino, R., Martinelli, L., Rastetter, E. B. and Sprent, J. I.: Towards an ecological understanding of biological nitrogen fixation, Biogeochemistry, 57–58(McKey 1994), 1–45, doi:10.1023/A:1015798428743, 2002.

Additionally, for the purposes of this study, the source of the DIN in the river is less important than the fact that there are large differences in NH4 concentration in the presence versus absence of effluent. In the stable isotope mixing model calculations we assume that phytoplankton in the +EFF parcel use NH4 or NO3 with no contribution from nitrogen fixation. We feel that this is a valid assumption given that a) this environment is unlikely to foster nitrogen fixation in general, and b) the +EFF plume (where we used the mixing model) would be even less likely to support nitrogen fixation given the elevated NH4 concentration.

[Figure]

**Fig. 1.** Dissolved oxygen in parcels with effluent and without effluent. Figure reproduced from figure S3D in Kraus et al. 2017

---

## Author Comment (AC2) · 14 Sep 2017

The manuscript by Schmidt et al. provides an interesting case study of the application of a very promising and little-used technique, by using flow cytometry as a sample preparation step to analyse d15N of phytoplankton to study nitrogen cycling in river systems . The ms is well written and data interpretation is generally solid.

I have relatively minor suggestions for improvement:

-While the focus of the manuscript is evidently on nitrogen cycling, it might be interesting to also report and discuss d13C values for the flow cytometry sorted phytoplankton samples – I assume these were analyses in the same run ?

AC:Unfortunately we were not able to measure $\delta$13C on the sorted samples. The peak focusing required to measure $\delta$15N for these small samples saturated the $\delta$13C signal. For this particular study, where we had a very limited amount of sample, we prioritized running duplicates for $\delta$15N over analyzing $\delta$13C. As described at the end of the manuscript, it would be ideal to change the field sampling approach in future studies to allow for sorting of more material. This should allow for isotopic analysis of specific populations within POM and it would also allow the method to be optimized for $\delta$13C analysis.

-the authors sampled following a Lagrangian approach – but it should perhaps be mentioned somewhere that the residence time of particles in a river system is expected to be higher than the water travel time in a river system, and some discussion on how that might affect the interpretation of results.

AC:We understand this concern. A single value for parcel travel time is best considered as an average residence time that represents a distribution of particle travel times. Because the focus of the study is on comparison of +EFF and –EFF conditions we took care to track parcel locations using surface current-following drifters, which track the velocity of a neutrally buoyant particles, and we also verified our position within each parcel by measuring changes in water quality that could be attributed to presence and absence of effluent. Since the parcels were ∼15 km long, we feel confident that our data represents particles exposed to these two different conditions even if we have slightly underestimated or overestimated the travel time of some particles.

This will be further clarified in the manuscript with the following text:

". . .field sampling was conducted using a Lagrangian sampling approach during October 24 to 29, 2013, and May 30 to June 4, 2014 (hereafter referred to as the "October"

and "June" experiments). During both October and June, sampling was coordinated with ∼20 hour WWTP effluent discharge holds, creating a ∼15 km stretch of effluent-free river to allow comparison of two parcels of river water; one containing effluent high in NH4+ (+EFF) and one without effluent (-EFF). During both experiments, +EFF and -EFF parcels were tracked using small drifters and a high-speed mapping boat equipped with a custom designed flow-through instrument package that continuously displayed surface-water measurements of specific conductance (a conservative tracer), to assure that samples were collected from within the same parcel of water as it travelled ∼70 km downstream (Fig. 1)."

-page 5, line 7: I assume this should be 0.7 _m GFF filters (not 70 _m) ?

AC: Typo will be corrected, GFF filters had a nominal 0.7 $\mu$m pore size.

-Methods: while the authors refer to Polissar et al. (2008) for the 'micro-EA' setup, the latter does not use a GasBench as interface, and the authors provide no details on the trapping/focussing of the eluting gases. Some more details would be of interest to readers who wish to setup a similar configuration.

AC: We used the same approach for trapping and focusing nitrogen as described in Pollisar et al. 2008, the use of the gasbench interface only allowed for automation of the trapping procedure.

This will be further clarified in the methods (pg 6 lines 24-32) with the following text,

"Sorted cells were transferred to 20 mL glass vials and dried down under vacuum using a centrifugal evaporator. Dried phytoplankton samples were redissolved in 20 $\mu$L ultra high purity deionized water and transferred into tin capsules. Capsules were dried overnight at 60 °C and then crushed into small cubes. $\delta$15N analysis of sorted phytoplankton was conducted using elements of a coupled Carlo Erba CHNS-O EA1108-Elemental Analyzer and Thermo Finnigan Gasbench II system with automated cryo-trapping system that is connected to an isotope ratio mass spectrometer (Thermo

Fisher Scientific) at the UC Santa Cruz Stable Isotope Laboratory facility. The elemental analyzer and gas bench were configured to run small samples using the methods described in Polissar et al. (2009). In this configuration, samples as small as 35 nmol N could be analyzed with a precision of 0.5 ‰. Phytoplankton samples analyzed for this study ($\delta$15N-PHY) ranged in size between 50 and 100 nmol N. Analysis of duplicate samples (sorted and analyzed independently) indicated a precision of 0.8 ‰ for the entire method.

-The nutrient concentration profiles clearly suggest that nitrification could be an important process affecting nutrient cycling in this system – it would be worth discussing this aspect and checking if there are data that might shed some light on this. Are there dissolved oxygen data available ? Have others measured nitrification rates in this system?

AC: Yes, other authors have investigated nitrification in the Sacramento River and we do discuss this and include reference to this literature on page 8, lines 14-19:

"Due to low concentrations of NH4+ upstream of the WWTP, it was only possible to measure $\delta$15N-NH4+ in the +EFF parcels downstream of the WWTP. In the +EFF parcels $\delta$15N-NH4+ increased from 7.9 ‰ to 9.7 ‰ in October and from 8.0 ‰ to 10.7 ‰ in June with downstream travel. We also observed that $\delta$15N-NH4+ increased while NO3- concentration increased and $\delta$15N-NO3 values decreased during transit in parcels containing effluent, which suggests nitrification was occurring. This observation is consistent with high rates of nitrification previously reported in the Sacramento River (Hager and Schemel, 1992; Parker et al., 2012a; O'Donnell 2014; Damashek et al., 2016)."

This is also a timely question as a paper estimating nitrification rates in this portion of the Sacramento River was just accepted for publication in Water Resources Research. We will add this additional reference to the manuscript:

Kraus, T.E.C. K. O'Donnell, B.D. Downing, J.R. Burau, B.A. Bergamaschi, in press.

Using paired in situ high frequency nitrate measurements to better understand controls on nitrate concentrations and estimate nitrification rates in a wastewater impacted river. Water Resources Research.

---

## Author Response (AR1)

**General comment:**

This study aims to trace the dissolved inorganic nitrogen source primarily used by phytoplankton in a river impacted by an heavy anthropogenic nutrient (ammonium) enrichment. The authors report an interesting dataset of stable nitrogen isotope ratio measurement in several inorganic and organic nitrogen pool. Moreover, they used a novel and elegant method (combination of flow cytometry cell sorting with stable isotope analysis) in order to distinguish (healthy) phytoplankton cells from the bulk particulate organic matter. The manuscript is well-written and the results reported by the authors are appropriately discussed, overall I greatly enjoyed reading this manuscript.

**Specific comment:**

*p4, line 26 : No results from a 15N-labeled nutrient uptake experiment are described in this manuscript, hence I would suggest to remove this from the material & method section.

AC: This text has been removed.

*p5, line 1-5 : Please, provide more information about the methodologies used to measure NO2-, NO3- and NH4+ concentration (the chemistry behind).

AC: The methods used to determine nutrient concentrations are described in Parker et al., 2012$a$, and this reference will be added to the text. The methods employed are also described below.

Nitrite concentration was determined by the Greiss reaction in which sulfanilamide and N-(1-Naphthy1) ethylenediamine dihydrochloride (NNED) reacts with nitrite in aqueous acidic solution to form an intensely pink diazo dye with an absorption maximum at 540 nm (Bendschneider and Robinson, 1952). To determine nitrate concentration, nitrate was reduced to nitrite by passage through a column containing copperized cadmium filings (Wood, Armstrong and Richards, 1967), and then the concentration of nitrate plus nitrite was determined in the manner described above. $NH_4^+$ concentrations were measured spectrophotometrically using a 10-cm path length cell according to the phenolhypochlorite method described in Solorzano (1969).

*p5, line 6 : I don't understand to what "70_m" is related. As you certainly know, nominal pore size of of the GFF filters is 0.7 _m. typo ?

AC: Typo corrected, GFF filters had a nominal 0.7 µm pore size.

*p8, line 6-19 : I would suggest to plot the d15N-NO3- and d15N-NH4+ data in a way similar to figure 3 (ie. Data plotted against travel time). These data are interesting, but it is difficult to visualize the trend when looking at table 1 only.

AC: $\delta^{15}$N-NO$_3^-$ and $\delta^{15}$N-NH4$^+$ are in fact plotted against travel time in Figure 5. A reference to figure 5 will be added to the

10  paragraph to improve clarity.

*p9, line 25 : typo : fluorescence, and not florescence.

AC: Typo corrected.

*p10, line 2 : Higher importance of labile POM mineralization in the +EFF parcel seems indeed plausible. It is a bit unfortunate that you did not measure heterotrophic bacteria abundance, but did you measure dissolved oxygen concentration, or any other variable related to ecosystem metabolism (for instance, community respiration)? They might be helpful to directly put in evidence a putative higher importance of heterotrophic metabolism in the +EFF parcels.

AC: Thank you for the helpful suggestion. We did measure dissolved oxygen (DO) concentration during the study, which reveals that DO was consistently lower in the parcels containing effluent (see figure S3D from Kraus et al. 2017$a$, included below), thus providing additional support for the argument that heterotrophic metabolism differed between +EFF and –EFF parcels.

This point has be clarified in the manuscript by adding the following text:

"It is also possible that the presence of effluent caused an increase in heterotrophy by bacteria and zooplankton in the +EFF parcels, resulting in $\delta^{15}$N-POM values greater than $\delta^{15}$N-PHY downstream of the WWTP. Heterotrophic bacterial abundance

30  was not measured during this study, but dissolved oxygen concentration was monitored, and in both October and June experiments dissolved oxygen concentrations were lower in the parcels containing effluent (Kraus et al., 2017$a$).

[Figure]

*p10, line 16 and below : I understood reading your paper that the diatom "health status" was decreasing downstream, then could it be hypothesized that a change in the composition of the phytoplankton assemblage downstream of the location where the effluent enter the Sacramento river explains the gradual increase in the contribution of NH4+ ? Did you look at the phytoplankton composition (and assess its variability) at several location during the travel of the two parcels of water downstream?

AC: This isotopic study was conducted in parallel with a larger investigation into this very important question and the results are reported in Kraus et al. 2017a. To summarize: Whole-water samples for phytoplankton enumeration were collected at each sampling point during downstream travel (15 points in October, 9 points in June). Patterns in the phytoplankton assemblages were examined using nonmetric multidimensional scaling (NMDS) ordinations constructed from Bray–Curtis similarity matrices (square root- transformed abundance data). Potential differences between the +EFF and -EFF parcels were tested for significance using analysis of similarity (ANOSIM). The results of this analysis showed a significant separation of phytoplankton assemblages between October and June, but no statistical difference between +EFF and –EFF parcels in October or June.

Given this finding we concluded it was unlikely that changes in community composition cause the greater increase in %NH4 in +EFF parcels compared to –EFF parcels. To clarify this point, we added the following sentence to the discussion section on pg 9 line 23:

"Patterns in the phytoplankton assemblages were examined and potential differences between the +EFF and -EFF parcels were tested for significance using analysis of similarity (ANOSIM). The results of this analysis showed no statistical difference between the assemblages present in +EFF and –EFF parcels in neither October nor June (Kraus et al. 2017$a$)."

*p10, line 16 and below : Beside NH4+ and NO3-, N2 fixation could also be a significant N source in systems where cyanobacteria are abundant. Do you know what was the contribution of cyanobacteria to the phytoplankton assemblage ? Could you explain why you rule out any contribution of N2 fixation as a N source in the Sacramento river?

AC: We are not aware of any studies that have looked at $N_2$ fixation in the Sacramento River. However, it is generally believed that $N_2$ fixation is not a significant process in large rivers like the Sacramento River, which is a deep (4-10 m), swift moving, channelized river lined with rip rap, which also has high concentrations of DIN. We consulted with several colleagues on this topic. For example Vitousek et al. 2002 states "The few studies of nitrogen fixation in flowing water

15  systems suggest that cyanobacteria and N fixation are limited by some of the same factors limiting other lotic algae: low light in small, heavily shaded streams (Horne & Carmiggelt 1975) and high current velocity (small to mid-sized streams) or high turbidity (large rivers). Indeed, planktonic cyanobacteria do not occur at all in most streams."

Vitousek, P. M., Cassman, K., Cleveland, C., Crews, T., Field, C. B., Grimm, N. B., Howarth, R. W., Marino, R., Martinelli,

20  L., Rastetter, E. B. and Sprent, J. I.: Towards an ecological understanding of biological nitrogen fixation, Biogeochemistry, 57–58(McKey 1994), 1–45, doi:10.1023/A:1015798428743, 2002.

Additionally, for the purposes of this study, the source of all of the DIN in the river is less important than the fact that there are large differences in $NH_4$ concentration in the presence versus absence of effluent. In the stable isotope mixing model

25  calculations we assume that phytoplankton in the +EFF parcel use $NH_4$ or $NO_3$ and we do not distinguish between $NH_4$ that originates in WWTP effluent or the sediment.

**Anonymous Referee #2**

The manuscript by Schmidt et al. provides an interesting case study of the application of a very promising and little-used technique, by using flow cytometry as a sample preparation step to analyse d15N of phytoplankton to study nitrogen cycling in river systems . The ms is well written and data interpretation is generally solid.

I have relatively minor suggestions for improvement:

-While the focus of the manuscript is evidently on nitrogen cycling, it might be interesting to also report and discuss d13C values for the flow cytometry sorted phytoplankton samples – I assume these were analyses in the same run ?

AC: Unfortunately we were not able to measure $\delta^{13}C$ on the sorted samples. The peak focusing required to measure a $\delta^{15}N$ for these small samples saturated the $\delta^{13}C$ signal.  For this particular study, where we had a very limited amount of sample, we prioritized running duplicates for $\delta^{15}N$ over analyzing $\delta^{13}C$.  As described at the end of the manuscript, it would be ideal to change the field sampling approach in future studies to allow for sorting of more material. This should allow for isotopic analysis of specific populations within POM and it would also allow the method to be optimized for $\delta^{13}C$ analysis.

-the authors sampled following a Lagrangian approach – but it should perhaps be mentioned somewhere that the residence time of particles in a river system is expected to be higher than the water travel time in a river system, and some discussion on how that might affect the interpretation of results.

AC: We understand this concern. A single value for parcel travel time is best considered as an average residence time that represents a distribution of particle travel times. Because the focus of the study is on comparison of +EFF and –EFF conditions we took care to track parcel locations using surface current-following drifters, which track the velocity of a neutrally buoyant particles, and we also verified our position within each parcel by measuring changes in water quality that could be attributed to presence and absence of effluent. Since the parcels were ~15 km long, we feel confident that our data represents particles exposed to these two different conditions even if we have slightly underestimated or overestimated the travel time of some particles.

This was further clarified in the manuscript with the following text:

"…field sampling was conducted using a Lagrangian sampling approach during October 24 to 29, 2013, and May 30 to June 4, 2014 (hereafter referred to as the "October" and "June" experiments). During both October and June, sampling was coordinated with ~20 hour WWTP effluent discharge holds, creating a ~15 km stretch of effluent-free river to allow comparison of two parcels of river water; one containing effluent high in $NH_4^+$ (+EFF) and one without effluent (-EFF). During both experiments, +EFF and -EFF parcels were tracked using small drifters and a high-speed mapping boat equipped with a custom designed flow-through instrument package that continuously displayed surface-water measurements of specific conductance (a conservative tracer), to assure that samples were collected from within the same parcel of water as it travelled ~70 km downstream (Fig. 1)."

-page 5, line 7: I assume this should be 0.7 _m GFF filters (not 70 _m) ?

AC: Typo corrected, GFF filters had a nominal 0,7 µm pore size.

-Methods: while the authors refer to Polissar et al. (2008) for the 'micro-EA' setup, the latter does not use a GasBench as interface, and the authors provide no details on the trapping/focussing of the eluting gases. Some more details would be of interest to readers who wish to setup a similar configuration.

AC: We used the same approach for trapping and focusing nitrogen as described in in Pollisar et al. 2008, the use of the gasbench interface only allowed for automation of the trapping procedure.

This was further clarified in the methods (pg 6 lines 24-32) with the following text,

"Sorted cells were transferred to 20 mL glass vials and dried down under vacuum using a centrifugal evaporator. Dried phytoplankton samples were redissolved in 20 µL ultra high purity deionized water and transferred into tin capsules. Capsules were dried overnight at 60 °C and then crushed into small cubes. $\delta^{15}N$ analysis of sorted phytoplankton was conducted using elements of a coupled Carlo Erba CHNS-O EA1108-Elemental Analyzer and Thermo Finnigan Gasbench II system with automated cryo-trapping system that is connected to an isotope ratio mass spectrometer (Thermo Fisher Scientific) at the UC Santa Cruz Stable Isotope Laboratory facility. The elemental analyzer and gas bench were configured to run small samples using the methods described in Polissar et al. (2009). In this configuration, samples as small as 35 nmol N could be analyzed with a precision of 0.5 ‰. Phytoplankton samples analyzed for this study ($\delta^{15}$N-PHY) ranged in size between 50 and 100 nmol N. Analysis of duplicate samples (sorted and analyzed independently) indicated a precision of 0.8 ‰ for the entire method.

-The nutrient concentration profiles clearly suggest that nitrification could be an important process affecting nutrient cycling in this system – it would be worth discussing this aspect and checking if there are data that might shed some light on this. Are there dissolved oxygen data available? Have others measured nitrification rates in this system?

AC: Yes, other authors have also investigated nitrification in the Sacramento River and we do discuss this and include reference to this literature on page 8, lines 14-19:

"Due to low concentrations of $NH_4^+$ upstream of the WWTP, it was only possible to measure $\delta^{15}$N-$NH_4^+$ in the +EFF parcels downstream of the WWTP. In the +EFF parcels $\delta^{15}$N-$NH_4^+$ increased from 7.9‰ to 9.7‰ in October and from 8.0 ‰ to 10.7

‰ in June with downstream travel. We also observed that $\delta^{15}$N-NH$_4^+$ increased while NO$_3^-$ concentration increased and $\delta^{15}$N-NO$_3$ values decreased during transit in parcels containing effluent, which suggests nitrification was occurring. This observation is consistent with high rates of nitrification previously reported in the Sacramento River (Hager and Schemel, 1992; Parker et al., 2012*a*; O'Donnell 2014; Damashek et al., 2016)."

This is also a timely question as a paper estimating nitrification rates in this portion of the Sacramento River was just accepted for publication in Water Resources Research. We will add this additional reference to the manuscript:

[revised manuscript text omitted]